# Laser-Induced Graphene Formation on Polyimide Using UV to Mid-Infrared Laser Radiation

**DOI:** 10.3390/polym15214229

**Published:** 2023-10-26

**Authors:** Vitalij Fiodorov, Romualdas Trusovas, Zenius Mockus, Karolis Ratautas, Gediminas Račiukaitis

**Affiliations:** 1Department of Laser Technologies, Center for Physical Sciences and Technology, Savanoriu Ave. 231, LT-02300 Vilnius, Lithuania; romualdas.trusovas@ftmc.lt (R.T.); karolis.ratautas@ftmc.lt (K.R.); g.raciukaitis@ftmc.lt (G.R.); 2Department of Chemical Engineering and Technology, Center for Physical Sciences and Technology, Sauletekio Ave. 3, LT-10257 Vilnius, Lithuania; zenius.mockus@ftmc.lt

**Keywords:** laser-induced graphene, galvanic plating, polyimide, polymers, deposition, selective, MID, copper

## Abstract

Our study presents laser-assisted methods to produce conductive graphene layers on the polymer surface. Specimens were treated using two different lasers at ambient and nitrogen atmospheres. A solid-state picosecond laser generating 355 nm, 532 nm, or 1064 nm wavelengths and a CO_2_ laser generating mid-infrared 10.6 µm wavelength radiation operating in a pulsed regime were used in experiments. Sheet resistance measurements and microscopic analysis of treated sample surfaces were made. The chemical structure of laser-treated surfaces was investigated using Raman spectroscopy, and it showed the formation of high-quality few-layer graphene structures on the PI surface. The intensity ratios I(2D)/I(G) and I(D)/I(G) of samples treated with 1064 nm wavelength in nitrogen atmosphere were 0.81 and 0.46, respectively. After laser treatment, a conductive laser-induced graphene layer with a sheet resistance as low as 5 Ω was formed. Further, copper layers with a thickness of 3–10 µm were deposited on laser-formed graphene using a galvanic plating. The techniques of forming a conductive graphene layer on a polymer surface have a great perspective in many fields, especially in advanced electronic applications to fabricate copper tracks on 3D materials.

## 1. Introduction

Over the past decade, graphene has demonstrated significant potential in the electronics industry due to its exceptional physical and chemical properties [1,2,3,4]. Given its advantageous properties, graphene finds applications across various fields, including automotive, medicine, and telecommunications. Previous research has demonstrated the utility of graphene in supercapacitors [5,6,7], sensors [8,9], batteries [10,11], optoelectronic devices [12,13,14], and photodetectors [15,16,17].

Various techniques could be employed to synthesise graphene, including mechanical and chemical methods. Mechanical methods involve peeling off graphite layers using ultra-sharp diamond tools [18,19] to create submicron-thick graphene layers or grinding graphite between balls until graphene flakes are formed to achieve nano-thickness [20,21]. The quality of graphene can be controlled by regulating process parameters such as grinding material or process time. However, the main disadvantages of mechanical methods are extended production time and repeatability issues, making this method less suitable for mass production. Chemical methods involve chemical vapour deposition when hydrocarbon gases such as methane and acetylene are decomposed to form graphene on a metallic catalyst. This method can produce even a single layer of graphene [22,23,24]. Alternatively, graphene can be prepared by epitaxial growth, where a hexagonal substrate such as silicon carbide is subjected to high temperature in a vacuum or inert atmosphere until the silicon sublimates, leaving behind carbon atoms to form graphene [25,26,27]. High-quality graphene could be produced by chemical methods, but it is expensive for mass production.

In 2014, it was found that graphene could be formed directly on some materials, such as polyimide (PI) films, after laser treatment [28]. The mechanism responsible for the formation of laser-induced graphene (LIG) varies depending on the wavelength of the laser used. In the case of infrared laser radiation, the photothermal effect has been proposed to account for the transition. This effect causes the instantaneous pyrolysis of the precursor, resulting in the breaking and recombination of chemical bonds with the release of gas [29]. On the other hand, the ultraviolet laser is more likely to cause a photochemical process. This is because the photon energy of the ultraviolet laser is close to that of chemical bonds, allowing it to directly break the chemical bonds in the precursor and generate LIG [30,31,32,33]. In the case of visible laser, both photothermal and photochemical effects contribute to LIG formation [34,35,36]. 

LIG formed on polymers could be used in electronic applications, such as for the selective fabrication of metallic tracks for interconnects [37]. This technology consists of two steps: the circuit pattern is laser-written on the polymer surface to form conductive graphene, and then the sample is immersed into the galvanic solution, where the laser-treated areas are deposited by copper or other metal. The technology could be used for the cost-effective production of copper tracks on 3D materials, where conventional methods are unsuitable. Another electronic application of LIG is for producing supercapacitors [38]. The graphene layer is first formed on the polyimide surface during this process. Then, it is stacked with a solid-state polymeric electrolyte and another laser-treated polyimide sheet. LIG supercapacitors have enhanced electrochemical performance and flexibility. Further, LIG application could be used in the production of gas sensors [39]. Laser-formed graphene on a polyimide sample could be used for gas sensing based on thermal conductivity. The electrical current passed through this type of sensor releases heat generated due to electrical resistivity. This excess energy is then dispersed into the surrounding environment, and the composition of the surrounding gases can be determined by measuring the resultant temperature change. These sensors could be used for sensing various gases and could be incorporated into many devices [40,41,42].

We report novel laser patterning techniques to produce a conductive graphene layer on the polymer surface to facilitate the metal deposition onto the formed graphene layer, making it suitable for various electronics applications. The main advantage is the ability to form conductive tracks on 3D materials, where conventional methods, like photolithography, are unsuitable. The method could be used for production of moulded interconnect devices (MIDs) or flexible electronics devices. Raman spectroscopy, scanning electron microscopy, and sheet resistance measurements were used to analyse formed graphene properties.

## 2. Materials and Methods

### 2.1. Materials

In our experiments, polyimide Kapton^®^ films manufactured by DuPont with a thickness of 127 µm were used. All samples were cleaned before the experiments using ethanol (obtained from Sigma-Aldrich, Burlington, MA, USA).

### 2.2. Laser Writing Techniques

Two different laser sources were used for the polyimide treatment. First was the solid-state picosecond laser Atlantic (Ekspla, Vilnius, Lithuania), generating 355 nm, 532 nm, and 1064 nm wavelengths. The pulse duration was 10 picoseconds, the pulse repetition rate was tuneable from 100 to 500 kHz, and the average laser power was up to 8.5 W. Another laser used in the experiment was pulsed CO_2_ laser Diamond E-150 (Coherent, Inc., Santa Clara, CA, USA), generating up to 150 W of average power at a 10.6 µm wavelength. The SCANgine 14 galvoscanner (Scanlab, Puchheim, Germany) was used for laser beam positioning. The processing speed varied from 30 mm/s to 200 mm/s, and laser line overlap (hatching) was changed from 0.01 mm to 0.1 mm. Four different f-theta objectives were used, with focal lengths of 100 mm (for 532 nm), 160 mm (for 1064 nm), 174 mm (for 355 nm), and 250 mm (for 10.6 µm) for laser beam focusing. The laser fluence was calculated by dividing the pulse energy by the area of the beam spot at 1/e^2^ level. The irradiation dose, which indicates the amount of laser energy absorbed, was calculated by multiplying the laser fluence by the number of laser pulses per beam spot area.

Several different treatment approaches were used to transform the polyimide surface layer into graphene (see Figure 1):Samples were treated with a defocused laser beam in an ambient atmosphere.Samples were treated with a defocused laser beam in a nitrogen atmosphere.Samples were pretreated at the focal plane and then repeatedly treated out of focal plane.Samples were dyed with black marker and treated with the defocused beam.

**Figure 1 polymers-15-04229-f001:**
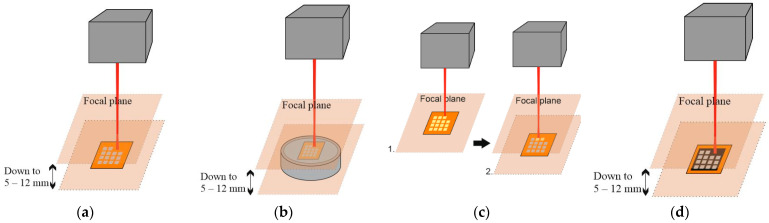
Polyimide surface processing techniques: (**a**) treatment with a defocused laser beam in an ambient atmosphere; (**b**) treatment with a defocused laser beam in a nitrogen atmosphere; (**c**) primary pretreatment (1) with a focused beam and secondary treatment (2) with a defocused laser beam; (**d**) treatment on a dyed surface with a defocused laser beam.

The optimal fabrication parameters were found by fabricating 10 × 20 mm^2^ rectangle areas, each time changing one of the fabrication parameters while keeping the others constant. 

Our previous research [37] showed that LIG is formed on the PI sample when it is placed below the focal plane and treated with the defocused laser beam. For LIG formation on the PI surface, it is not only essential to select the laser fluence properly; the exposure time is also crucial. The treatment out of focus allows the control of laser fluence, laser spot size, and pulse overlap on the polyimide surface. This way, suitable exposure time and surface temperature could be preserved to facilitate graphene formation. The proper sample distance to the focal plane to transform PI into LIG could be roughly found after an inspection of the PI sample, which was treated at a 45-degree angle (Figure 2). 

The sheet resistance and Raman spectra of LIG structures formed on a polyimide surface using various laser treatment parameters were measured and compared in our study.

### 2.3. Sheet Resistance Measurements

The study conducted sheet resistance measurements of laser-processed regions using the four-probe method, which involved four probes arranged in a linear order with a 1 mm gap between them. The external probes served as the current source, while the inner probes measured the voltage. The acquired voltage and current measurements enabled the calculation of the surface resistance. Sheet resistance measurements were performed using the source meter (Keithley 2602A, Keithley, Cleveland, OH, USA).

### 2.4. Surface Morphology Analysis

Microscopic observation of the fabricated sample surface was performed using an optical microscope (Olympus BX52, Olympus, Tokyo, Japan) with a CCD camera. Objectives with magnifications ranging from 5× to 50× were used to image samples in bright and dark field illumination modes. In addition, a scanning electron microscope (JSM-6490LV, JEOL Ltd., Tokyo, Japan) was used for surface analysis. Before SEM analysis, samples were gold-coated to avoid charging the polymer surface using a magnetron sputter coater (Q150T ES, Quorum Technologies Ltd., Laughton, UK).

### 2.5. Raman Analysis

Raman measurements were performed with the Raman spectrometer/microscope in Via (Renishaw, UK) equipped with the thermoelectrically cooled (−70 °C) CCD detector. Spectra were excited with the 532 nm wavelength laser and dispersed by 1800 grooves/mm grating. Raman spectra were taken using a 50×/0.75NA objective lens. 

## 3. Results and Discussion

### 3.1. Fabricating with 355 nm, 532 nm, and 1064 nm out of the Focal Plane

In the first part of the experiments, polyimide samples were treated with the defocused laser beam, using 355 nm, 532 nm, or 1064 nm wavelengths in the ambient atmosphere. The goal was to understand how each wavelength affects the formation of LIG. The photon energy of an ultraviolet laser is sufficient to directly break the bonds in polyimide, as the C–C bond energy is approximately equal to the photon energy corresponding to the 355 nm wavelength. Alternately, when infrared radiation is used, the photon energy is too low to break the chemical bonds, resulting in heating effects dominating. Using visible wavelength, both effects take part [28,29,30,31,32,33,34,35,36]. 

Samples were initially placed at the focal plane of the laser beam and then moved down below the focal plane up to 12 mm in 0.25 mm steps and processed with the defocused laser beam. All the parameters were kept constant except the sample distance to the focal plane. A fixed average laser power of 1.8 W was used with all three wavelengths, the pulse repetition rate was 100 kHz, and the scanning speed was 100 mm/s. Rectangles of size 10 × 20 mm^2^ were fabricated on the sample surface. 

The irradiation doses from 2.1 J/cm^2^ to 14.8 J/cm^2^ were needed to transform the polyimide surface into LIG structures treated with a 355 nm wavelength. LIG was found on the sample surface after irradiation from 3.2 J/cm^2^ to 14 J/cm^2^ with 532 nm wavelength and after irradiation from 4.4 J/cm^2^ to 10.9 J/cm^2^ with 1064 nm wavelength (see Figure 3). The lowest sheet resistance of processed sample areas where LIG appeared was 18 Ω, treating with 355 nm, 13 Ω with 532 nm, and 8 Ω with 1064 nm. Fabrication with 355 nm and 532 nm required a lower irradiation dose to form LIG than with 1064 nm. 

When the sample was placed at a distance less than 4 mm below the focal plane, in other words, treatment with a highly focused beam (irradiation dose higher than 26 J/cm^2^), this led to overheating of the polyimide sample surface and formation of a black-coloured soot layer with a sharp increase in sheet resistance. A ticker soot layer with reduced sheet resistance was formed by reducing the beam’s diameter. The sheet resistance of the soot layer, formed after the treatment with 26 J/cm^2^ irradiation dose, was 20 MΩ, and at the focal plane it was 70 Ω.

The surface morphology of formed graphene layers, treated with 7 J/cm^2^ at 355 nm, 532 nm, and 1064 nm wavelengths, is shown in Figure 4. LIG, which was formed after irradiating with 1064 nm wavelength, was partially cracked. No surface cracks were observed on the samples treated with 355 nm and 532 nm radiations. LIG’s surface formed after irradiation with 355 nm and 532 nm and was smooth and well bonded to the polyimide substrate. It did not peel off after bending. This might be caused by different LIG formation mechanisms irradiating with different wavelengths: to form LIG with infrared radiation causes a heating effect and pyrolysis of the sample surface; on the other hand, when irradiating with ultraviolet, the photon energy is high enough to directly break a chemical bond of the material [28,29,30,31,32,33]. 

### 3.2. LIG Formation in the Nitrogen Atmosphere

The laser treatment mechanism of PI in the air can be described as the oxidative reaction between laser-irradiated PI and atmospheric oxygen. Atmospheric oxygen results in high-temperature degradation of the PI structure. In contrast, in treatment under inert atmospheric conditions, the absence of an oxidising agent leads to the formation of a comparatively stable graphene structure [43,44,45].

The polyimide sample was placed in a chamber with nitrogen, and the surface was treated with a defocused laser beam at the 1064 nm wavelength. The chamber was moved down from 0.25 mm to 8 mm below the focal plane. The pulse repetition rate was 100 kHz, and the sample processing speed was 100 mm/s. Surface modification occurred when the irradiation dose was 5 J/cm^2^ or higher. LIG with the lowest sheet resistance of 5 Ω was formed on the sample surface when the sample was placed from 7 to 8 mm below the focal plane corresponding to the irradiation dose from 7.3 J/cm^2^ to 7.8 J/cm^2^.

### 3.3. LIG formation on the Pretreated Area; LIG Formation on the Dyed Surface

To investigate the effect of polyimide surface absorption, we conducted experiments on both pretreated and dyed polyimide surface areas. We hypothesised that a lower irradiation dose would be needed to form graphene if the polyimide surface were to be made absorbent. 

At first, the PI sample was placed at the focal plane of the f-theta objective and treated with a highly focused 1064 nm beam at a repetition rate of 100 kHz, processing speed of 100 mm/s, and laser power was set to 0.8 W. This laser power was too low to form soot and affect the sheet resistance of the sample surface. However, a colour change occurred on the sample surface, and the polyimide surface area became brown-coloured after laser treatment. After that, the sample was lowered down to 10 mm below the focal plane, and the same area was treated again with a defocused 1064 nm beam, and only 2.3 J/cm^2^ of irradiation dose was required to form LIG on the surface (see Figure 5a). Forming LIG on a pretreated area required half the irradiation dose of forming on a surface without pretreatment. To compare, 4.4 J/cm^2^ was needed for formation of LIG on a not-pretreated PI surface. The sheet resistance of the formed LIG was 8 Ω.

In the following experiment, LIG was formed on the polyimide surface dyed with a black marker. The PI sample was placed 10 mm below the focal plane and treated with a defocused laser beam using 1064 nm wavelength, 100 kHz pulse repetition rate, and 100 mm/s scanning speed. The treatment on the pretreated area led to a lower irradiation dose required to form LIG than direct processing on a clean surface (see Figure 5b). Formation of LIG was initiated after treating PI with an irradiation dose of 2.1 J/cm^2^. To compare, treating a clean surface, placed 10 mm below the focal plane and using the same fabrication parameters, required a 4.4 J/cm^2^ irradiation dose. The sheet resistance of the formed structure was as low as 9 Ω.

### 3.4. PI Processing with 10.6 µm Wavelength

In the next part of the experiments, polyimide samples were processed with a CO_2_ laser, generated at the 10.6 µm wavelength. In this case, the energy carried by photons was significantly lower than the energy required to break C–C bonds. As a result, the heating effect had a substantial impact. We used the longest focal length objective (focal length of +250 mm) for sample treatment with the CO_2_ laser to compare with our previously described measurement. This led to the enlargement of the beam spot diameter, and due to this, the laser power was increased to 10 W to keep the same irradiation doses as in previously described treatments using 355 nm, 532 nm, and 1064 nm wavelengths. Samples were processed at a 100 kHz pulse repetition rate, and the scanning speed was set to 200 mm/s. The laser beam spot diameter at the focal plane was 400 µm, and the laser-line overlap hatching was 200 µm. Samples were treated with a defocused laser beam at different distances when samples were placed down as low as 42 mm below the focal plane in 3 mm steps. When treating at the focal plane or down to 6 mm below the focal plane, samples were burned through due to too-high laser intensity. The lowest sheet resistance was 7 Ω after fabrication when the sample was placed 9 mm below the focal plane with an irradiation dose of 8.6 J/cm^2^ (see Figure 6). 

### 3.5. Structural Analysis by Raman Spectroscopy

Raman spectra of graphene-like materials show standard features in the 1000–3000 cm^−1^ region. The G-peak at ~1560 cm^−1^ corresponds to the E_2g_ phonon in the Brillouin zone centre. The D-peak at ~1360 cm^−1^ is due to the breathing modes of sp^2^ atoms and requires a defect for its activation. Therefore, this peak shows the presence of structural defects in graphene. The 2D peak at ~2700 cm^−1^, the second order of the D-peak, is the most intrinsic to graphene [46]. The intensity ratio for the D, G, and 2D Raman bands is often used to evaluate the graphene phase quality. The minimum of I(D)/I(G) and a maximum of I(2D)/I(G) correspond to the highest quality of graphene.

Firstly, Raman spectroscopy measurements were performed on the samples, treated in an ambient atmosphere with a defocused laser beam at 355 nm, 532 nm, and 1064 nm wavelengths. The sheet resistance of samples treated with the defocused beam at 355 nm wavelength was 18 Ω, with 532 nm—13 Ω and 1064 nm—8 Ω. The intensity peak ratios, the well-known graphene quality parameters, were I(D)/I(G) = 1.05 and I(2D)/I(G) = 0.5 for samples treated with the 355 nm defocused laser beam at an irradiation dose of 7.5 J/cm^2^. I(D)/I(G) = 0.92 and I(2D)/I(G) = 0.65 occurred for samples treated with defocused 532 nm wavelength at an irradiation dose of 6.8 J/cm^2^ and I(D)/I(G) = 0.57 and I(2D)/I(G) = 0.58 occurred for samples treated with 1064 nm defocused laser beam at an irradiation dose of 7.3 J/cm^2^. The lowest I(D)/I(G) ratio and the highest I(2D)/I(G) Raman peaks were obtained by treating with defocused 1064 nm laser irradiation (Figure 7). 

Further, samples treated with a defocused 1064 nm wavelength beam in the nitrogen atmosphere were measured. During the treatment conducted within an inert atmosphere, the lack of an oxidising agent resulted in the generation of a relatively stable configuration of graphene. Treating samples with an irradiation dose of 7.3 J/cm^2^ and 7.8 J/cm^2^ in a nitrogen chamber led to the formation of the highest intensity 2D peaks in our experiment (Figure 8). The intensity ratios I(D)/I(G) and I(2D)/I(G) of samples treated with 7.3 J/cm^2^ were, accordingly, 0.46 and 0.81. The intensity ratios of samples treated with 7.8 J/cm^2^ were 0.6 and 0.81. The sheet resistance of samples treated in a nitrogen chamber with irradiation doses of 7.3–7.8 J/cm^2^ was the lowest, equal to 5 Ω.

Later, Raman spectra were measured on the samples treated with 10.6 µm wavelength when a sample was moved down to 42 mm from the focal plane in 3 mm steps. The highest intensity 2D peak was observed on the sample, which was treated at a 9 mm distance from the focal plane with an irradiation dose of 8.6 J/cm^2^, and the sheet resistance of the formed structure was 7 Ω. In this case, I(D)/I(G) = 0.57 and I(2D)/I(G) = 0.63. The results strongly correlated with sheet resistance measurements of the sample surfaces treated at various distances to the focal plane. Intensity ratios I(2D)/I(G) were highest when the sheet resistances of the treated sample surfaces were lowest, and the intensity ratio value decreased when sheet resistance increased (Figure 9).

Table 1 presents the lowest sheet resistances and best Raman spectra characteristics of the samples, treated with various laser wavelengths. The results showed that LIG with the lowest sheet resistance and best Raman spectra characteristics could be formed on the PI surface after irradiation with the defocused 1064 nm beam in the nitrogen atmosphere. Eliminating oxygen in an inert atmosphere laser treatment promoted stability in the graphene structure.

### 3.6. Metal Plating on LIG

The metal deposition on the LIG layer was performed using galvanic plating. Different LIG surface patterns formed after treatment with infrared and visible or ultraviolet wavelengths, as presented in Figure 4. LIG formed after the treatment with 355 nm or 532 nm radiation and was better bonded to the sample surface than with 1064 nm, 1064 nm treated in nitrogen atmosphere, or with 10.6 µm. The formed graphene layer could not be removed after bending or rubbing. Fabrication with infrared radiation led to the formation of low sheet resistance, but worse bonding to the polyimide surface. The graphene layer could be peeled off after some bending. Therefore, PI samples treated with 532 nm wavelength were selected for metal-plating experiments. 

Copper (Cu), silver (Ag), and nickel (Ni) were chosen for the galvanic plating. Copper and silver have the highest electrical conductivity of all the metals [47] and are widely used to produce electrical tracks in electronics. On the other hand, nickel is an excellent material for electroplating due to its compactness: nickel could be selectively plated on tiny structures [48]. Also, nickel is used as a sublayer for copper deposition in electronics to enhance adhesion [49,50]. Laser-treated 10 × 20 mm^2^ areas of PI film were connected to the cathode terminal, and sheets of metal to deposit were used as an anode. Deionised water was used for solutions production. Magnetic stirring of 550 rpm was used in all solutions during plating. Plating parameters are listed in Table 2.

In the first part of the experiment, galvanic copper plating was performed. The lowest copper thickness required for full coverage of the area of the formed 13 Ω LIG layer was 3 µm. On samples with a higher sheet resistance, a thicker layer of copper was required to obtain full coverage of the patterned area: 25 Ω—7 µm thickness copper layer, 30 Ω—10 µm thickness copper layer. 

We did not succeed in uniform, defect-free plating when the treated surface sheet resistance was higher than 100 Ω. The plating was uneven, with many undeposited areas, and plating was more intensive around the cathode terminal. An LIG layer with a low sheet resistance is needed to deposit a smooth metal layer. The lower the surface sheet resistance, the thinner and more continuous the metal layer that can be deposited (Figure 10).

Further, plating with nickel and silver was performed on LIG samples with a sheet resistance of 13 Ω. The lowest thickness nickel layer required for complete deposition of LIG was 1 µm. The lowest thickness when plating with silver was 9 µm. The thickness of the deposited metal layer depended not only on the surface sheet resistance but also on the type of metal to deposit.

## 4. Conclusions

In this research, the interaction between laser radiation and polyimide surface was examined, and various techniques were applied for treating a polyimide surface to form LIG. The LIG formation was compared on polyimide surfaces treated with ultraviolet (355 nm), visible (532 nm), and infrared (1064 nm and 10.6 µm) wavelengths. The results showed that an infrared laser was more suitable for forming a few-layer LIG than ultraviolet or visible lasers, as evidenced by lower defect band intensity in Raman spectra and lower sheet resistance on the LIG surface. The sheet resistance of the LIG structure was strongly correlated with the I(D)/I(G) and I(2D)/I(G) ratios. Furthermore, the morphology of the formed LIG surface differed when treated with infrared and visible or ultraviolet radiation. Irradiation with infrared resulted in a very smooth LIG surface with cracks, while treatment with visible and ultraviolet produced a rougher LIG surface without cracks. The highest quality LIG was obtained when the sample was treated with a defocused 1064 nm laser beam in a nitrogen atmosphere. Irradiation doses from 7.3 J/cm^2^ to 7.8 J/cm^2^ were used to form LIG with excellent surface sheet resistance of 5 Ω. Raman spectra of treated samples showed the highest intensity ratio I(2D)/I(G) = 0.81. 

Two novel techniques for LIG formation on PI surface were disclosed in our study. Polyimide surface treatment with a defocused laser beam on the pretreated area or surface treatment with a defocused laser beam on the dyed area to form LIG required a much lower irradiation dose than treatment on a clean surface. 

The quality of polyimide transformation to electroconductive LIG was investigated by galvanic deposition of metals on the LIG surface, and it was found that the thickness and uniformity of the deposited metal layer were correlated with the sheet resistance of the LIG surface. Thinner and more uniform metal layers were obtained on surfaces with a lower sheet resistance, and the type of metal also influenced the thickness of the deposited layer. Higher sheet resistance required thicker metal layers to achieve full coverage, and a uniform copper layer could not be deposited when sheet resistance was higher than 100 Ω. 

## Figures and Tables

**Figure 2 polymers-15-04229-f002:**
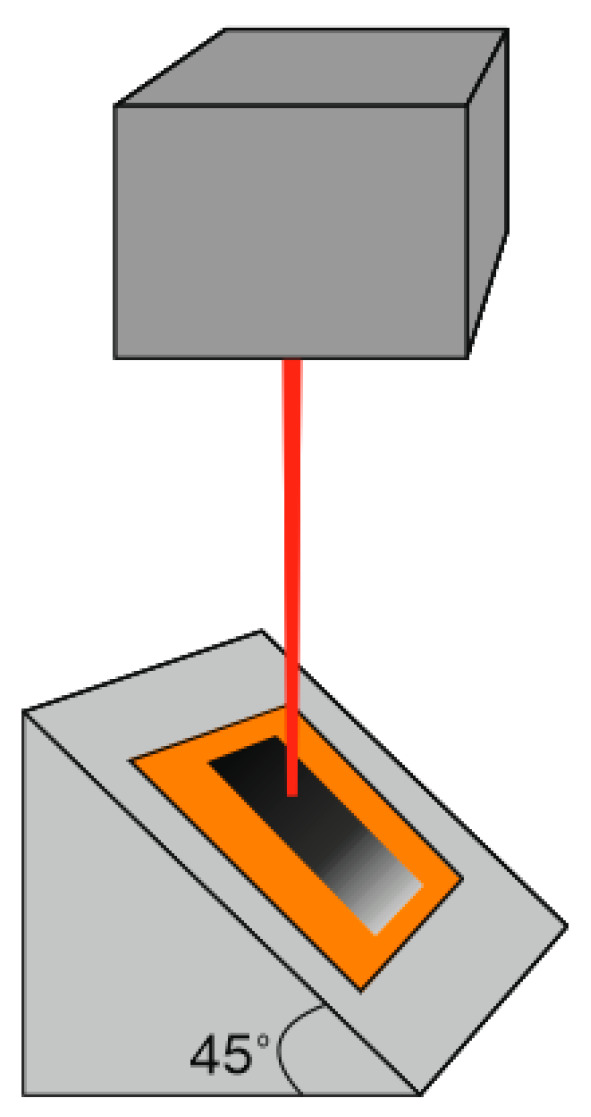
Polyimide sample, placed at a 45-degree angle and laser-treated. In this way, various areas of the sample are treated at various distances to the focal plane and a correct distance to the focal plane for high-quality LIG formation could be found.

**Figure 3 polymers-15-04229-f003:**
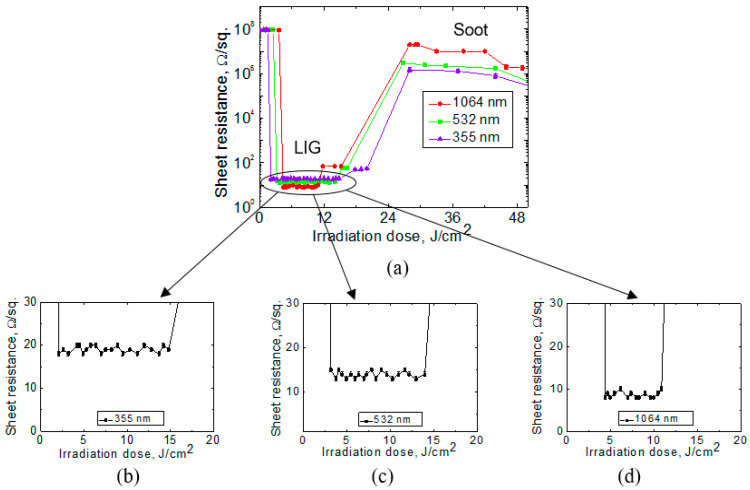
The sheet resistance of the laser-treated surface after irradiation with 355 nm, 532 nm, and 1064 nm wavelengths with irradiation doses up to 50 J/cm^2^ (**a**). Graphs (**b**–**d**) show magnified sheet resistance dependence on the irradiation dose where LIG was achieved.

**Figure 4 polymers-15-04229-f004:**
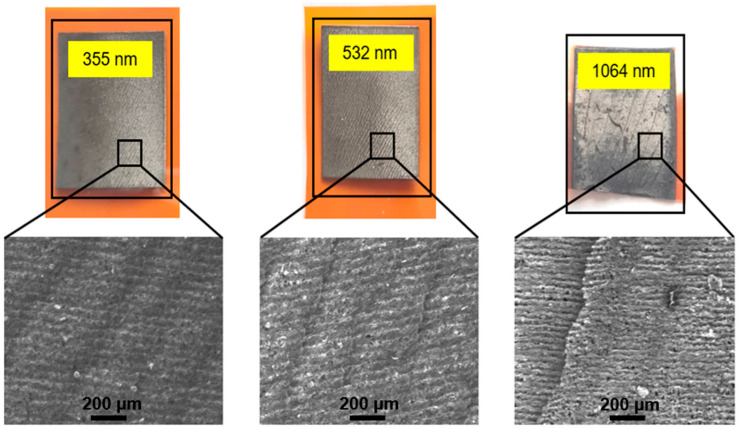
LIG formed on the Kapton PI surface after treatment with 7 J/cm^2^ irradiation dose using 355 nm, 532 nm, and 1064 nm wavelength. Surface cracks were found on LIG, formed with 1064 nm.

**Figure 5 polymers-15-04229-f005:**
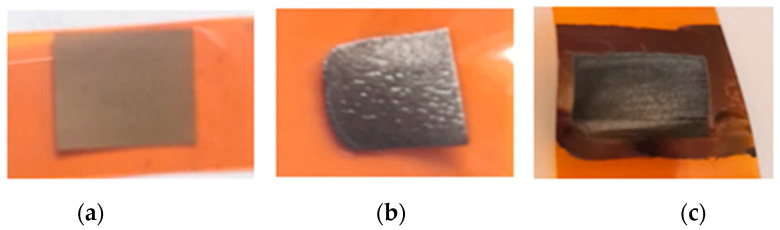
(**a**) Sample pretreated with a highly focused beam. Colour modification appeared on the PI surface after the pretreatment; (**b**) LIG formed after treatment with a defocused laser beam on the pretreated area; (**c**) LIG formed on a dyed polyimide surface with a defocused 1064 nm laser beam.

**Figure 6 polymers-15-04229-f006:**
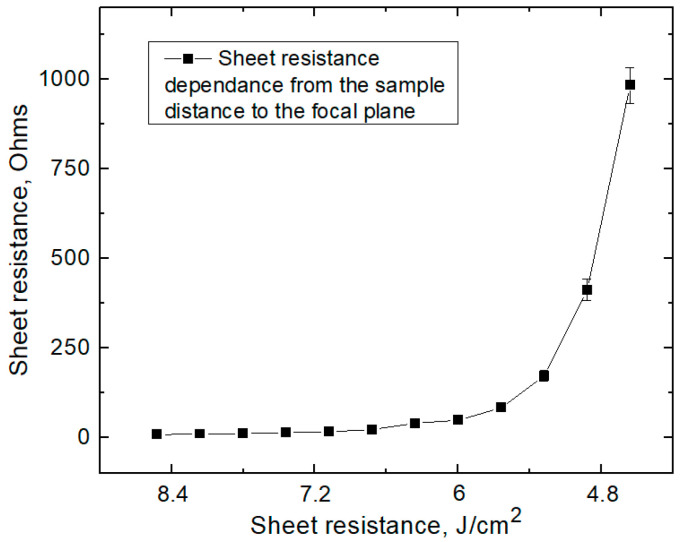
Treated surface sheet resistance after irradiation with 10.6 µm at different distances to the focal plane.

**Figure 7 polymers-15-04229-f007:**
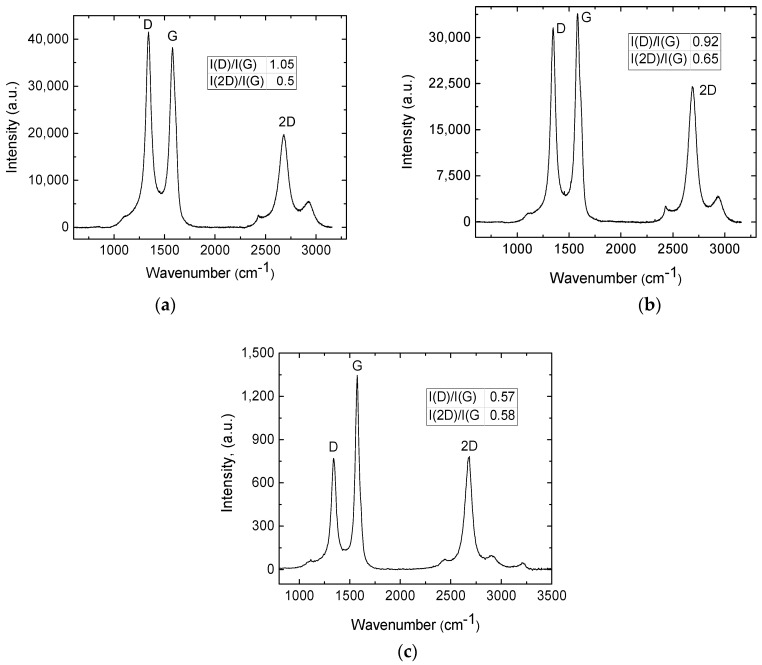
Raman spectra are shown of the sample treated with a defocused laser beam at 355 nm and irradiation dose 7.5 J/cm^2^ (**a**); 532 nm and irradiation dose 6.8 J/cm^2^ (**b**); and 1064 nm wavelength and irradiation dose 7.3 J/cm^2^ (**c**).

**Figure 8 polymers-15-04229-f008:**
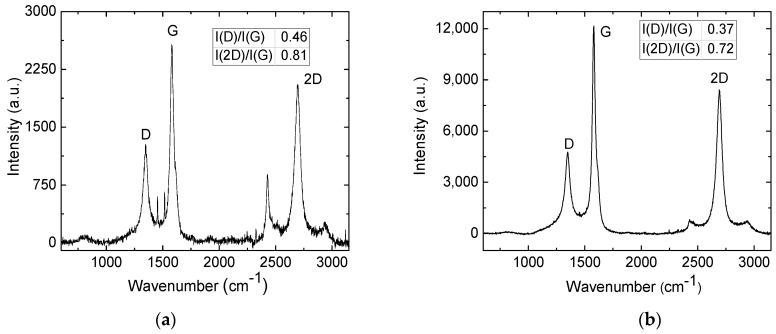
Raman spectra of the sample surface, treated with 1064 nm wavelength at 7 mm below the focal plane in the nitrogen atmosphere, treated with 7.3 J/cm^2^ (**a**) and 7.8 J/cm^2^ (**b**).

**Figure 9 polymers-15-04229-f009:**
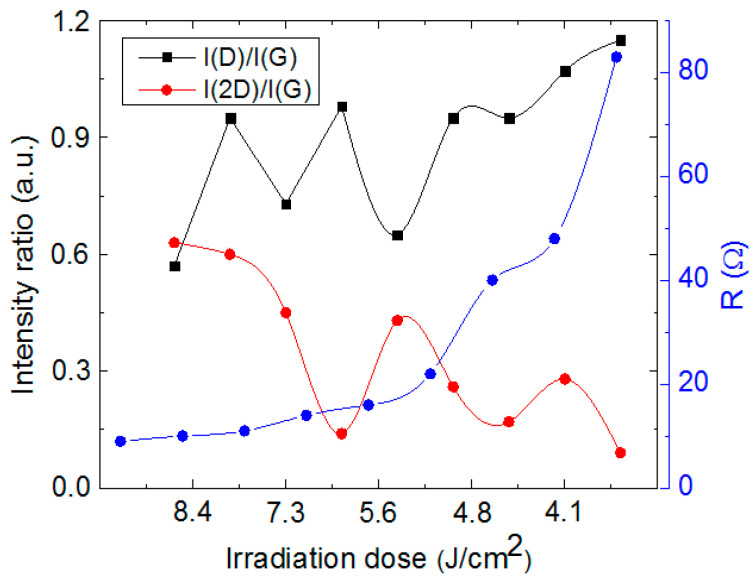
Sheet resistance dependence on the irradiation dose after a CO_2_ laser (blue line) treatment. Dependence of intensity ratio I(D)/I(G) (black line) and I(2D)/I(G) (red line) on the irradiation dose.

**Figure 10 polymers-15-04229-f010:**
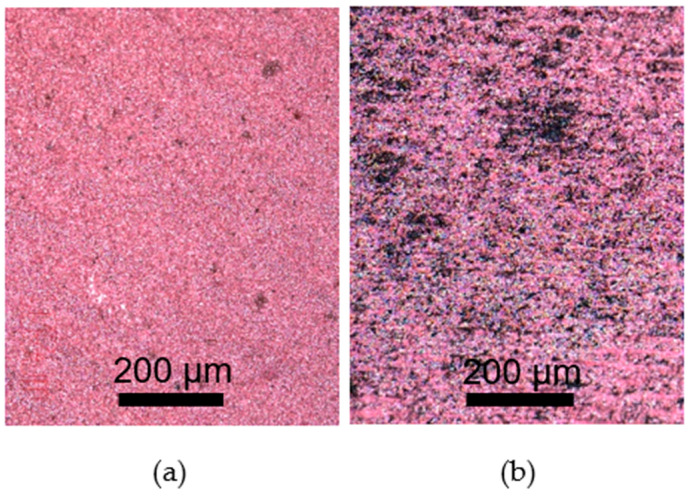
(**a**) A uniform 7 µm thickness copper layer, deposited on the LIG surface with a sheet resistance of 13 Ω; (**b**) copper deposited on the LIG surface with a sheet resistance of 100 Ω.

**Table 1 polymers-15-04229-t001:** Sheet resistance and Raman spectra parameters listed after treatment with different wavelengths.

Wavelength, nm	Irradiation Dose, J/cm^2^	Sheet Resistance, Ω	Adhesion	I(D)/I(G)	I(2D)/I(G)
355	7.5	18	Strong	1.05	0.5
532	6.8	13	Strong	0.92	0.65
1064	7.3	8	Weak	0.57	0.58
10,600	8.6	7	Weak	0.57	0.63
1064, in the nitrogen atmosphere	7.3	5	Weak	0.46	0.81
1064, in the nitrogen atmosphere	7.8	5	Weak	0.37	0.72

**Table 2 polymers-15-04229-t002:** Electroplating parameters for galvanic metal plating on LIG samples.

Metal	Solution	Current Density, mA/cm^2^	Plating Speed, µm/min
Cu	H_2_SO_4_ 50 g/L, CuSO_4_ ·H_2_O 200 g/L	20	0.44
Ni	NiSO_4_·7H_2_O 250 g/L, NiCl_2_·6H_2_O 45 g/L, H_3_BO_3_ 38 g/L	20	0.41
Ag	Commercial cyanide bath	4	0.256

## Data Availability

The data supporting this study’s findings are available from the corresponding author upon reasonable request.

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
