# Peer review of "Laser-Induced Graphene Formation on Polyimide Using UV to Mid-Infrared Laser Radiation"

_polymers, 2023, doi:10.3390/polym15214229_

Round 1

Reviewer 1 Report

1.line 107~108 , defocused lase beam must be revised in format

2.line 149, the title 2.4 should be revised to 2.5  

Author Response

Our team would like to thank the reviewer for given time to review the manuscript. The response to the comments is in attachment. All the revisions are highlighted in yellow.

Reviewer 2 Report

In order to produce conductive graphene layers for integrating circuit applications, polyimide surfaces in serie were pulsed by using lasers of different type and consequence wevelenght. Then, the polymer matrix structure was characterized by Raman Spectroscopy also evaluating the conductivity of the samples. Further, a copper layer was deposited on the surface of graphene by galvanic plating for realized integrate stable circuits.

As indicated below, some figures appear not clear 

Line 24. The keywords appear to be more general, it's indicated to introduce more terms.

Line 243. These figures appear to be not focused and clear.

The English form reported in the paper is adequate, only a minor revision is required.

Author Response

(The authors gave the same response as above.)

Reviewer 3 Report

The authors present a well thought out study of the formation of LIG on kapton.

I recommend publication after minor corrections of the following:

1)    Include an example in the introduction of where explicitly the method can have high impact. The authors allude to “the cost-effective production of copper tracks on 3D materials, where conventional methods are unsuitable” but no specific application is included.

2)    Figures: Improve labeling on SEM images in figure 4. Chose the same x-axis for figures 3 and 6. Either use distance or dose but please keep consistent for easier comparability.

3)    Line 228: “Forming LIG on a pre-treated area required almost twice the lower irradiation dose than forming on a surface without pretreatment. “ I believe this should read “…required HALF the dose…”

4)    Comment on why the lowest sheet resistance (5 Ω) LIG was not tested for electroplating.

Author Response

(The authors gave the same response as above.)
